# Detection of CADM1, MAL, and PAX1 Methylation by ddPCR for Triage of HPV-Positive Cervical Lesions

**DOI:** 10.3390/biomedicines13061450

**Published:** 2025-06-12

**Authors:** Maria Anisimova, Mark Jain, Liya Shcherbakova, Liana Aminova, Andrey Bugerenko, Natalia Novitskaya, Larisa Samokhodskaya, Vladislav Kokarev, Victoria Inokenteva, Olga Panina

**Affiliations:** 1Medical Research and Education Institute, Lomonosov Moscow State University, 119192 Moscow, Russia; jain-mark@outlook.com (M.J.); liya.fbm@gmail.com (L.S.); jeddit@yandex.ru (A.B.); nna2518208@yandex.ru (N.N.); slm@fbm.msu.ru (L.S.); kokarev.vladislav01@bk.ru (V.K.); inokentyeva_victoria@bk.ru (V.I.); olgapanina@yandex.ru (O.P.); 2Center Aviamed, 101000 Moscow, Russia; aminovaln22@gmail.com

**Keywords:** *CADM1*, cervical cancer, cervical intraepithelial neoplasia, DNA methylation, digital PCR, epigenetics, HPV, HSIL, *MAL*, *PAX1*

## Abstract

The aberrant DNA methylation of tumour suppressor genes, including *CADM1*, *MAL*, and *PAX1*, is implicated in cervical carcinogenesis. **Objectives:** This pilot study aimed to evaluate the methylation levels of these genes in HPV-positive women and assess their diagnostic performance for detecting histologic high-grade squamous intraepithelial lesions (HSILs) and carcinoma. **Methods:** Cervical samples from 73 HPV-positive women were analyzed using droplet digital PCR (ddPCR) to quantify methylation levels of *CADM1*, *MAL*, and *PAX1*. The methylation levels were further compared across cytological and histological classifications. A control group of 26 HPV-negative women with negative cytology was also included. The diagnostic performance was assessed through receiver operating characteristic (ROC) analysis, as well as sensitivity and specificity calculations for individual genes and gene panels. **Results:** *MAL* methylation was absent in NILM, LSIL, and HSIL samples but was significantly elevated in carcinoma. *PAX1* methylation was observed in both high-grade and some low-grade lesions. *CADM1* methylation remained low or undetectable in the NILM, LSIL, and HSIL groups, with a significant increase observed in carcinoma cases. The *CADM1/MAL* panel demonstrated the highest diagnostic accuracy, with an area under the curve (AUC) of 0.912, 70% sensitivity, and 100% specificity. ddPCR exhibited superior analytical sensitivity compared to real-time PCR. **Conclusions:** The *CADM1/MAL* methylation panel, assessed by ddPCR, may serve as a specific biomarker for the triage of HPV-positive women at risk of HSIL and carcinoma. However, this study’s limited sample size and single-centre design necessitate cautious interpretation. Further validation in larger, population-based cohorts is necessary to confirm its clinical utility.

## 1. Introduction

Cervical cancer (CC) ranks as the fourth most prevalent malignancy affecting women globally [1]. Researchers have recognized human papillomavirus (HPV) as a key factor in the development of CC [2,3]. This virus belongs to the Papillomaviridae family and is a non-enveloped virus with a circular, double-stranded DNA genome [4].

HPV classification is based on nucleotide sequence similarity in the open reading frame of the capsid protein L1 [5]. To date, 231 HPV types have been identified and are subdivided into Alpha-, Beta-, Gamma-, Mu-, and Nu-papillomaviruses [6]. Each genus of HPV has a specific tropism for different cells of the human body, determined by the interaction specificity between the protein L1 and the cell surface [7]. Researchers have confirmed the carcinogenic effects of fifteen types of HPV: 16, 18, 31, 33, 35, 39, 45, 51, 52, 56, 58, 59, 68, 73, and 82 [8]. Among these strains, HPV16 is the most predominant, accounting for 60% of cases [9]. HPV infects the basal layer of the stratified epithelium, typically through microtraumas or at the transition zone, where the squamous epithelium transforms into glandular tissue [10]. The virus replicates its genome in coordination with host cell division [11]. Although HPV testing has become an essential diagnostic tool, epidemiological studies reveal an interesting feature: HPV infection occurs far more frequently than cervical intraepithelial neoplasia (CIN) [12]. This finding indicates that the immune system can often clear the infection independently, with transient cervical changes frequently resolving within a year [13]. Therefore, HPV infection alone is insufficient to initiate carcinogenesis. Research has shown that the development of invasive cervical cancerous lesions is associated with both genetic and epigenetic alterations in the host and/or viral genome [14].

DNA hypermethylation is an early and frequent event in various types of cancer, and the methylation of viral DNA has recently been proposed as a new biomarker for cervical pathology [15,16]. With the advancement in sequencing technologies, the number of studies investigating the role of DNA methylation in disease progression has increased [17]. A current area of research focuses on identifying DNA methylation patterns that can help predict the outcome of persistent HPV infection and the development of CIN [18]. Researchers have thoroughly investigated methylation within the HPV genome to identify specific markers that can distinguish CIN from CC [17]. Studies have established a correlation between CIN/CC and the methylation of CpG sites in the HPV genome, including L1, L2, E2–E5, and URR [18]. It is essential to note that the positive correlation between L1 methylation levels and CIN2+ lesions has been confirmed by numerous studies [14]. Moreover, a meta-analysis of methylation, including 68 different genes from 51 studies, revealed that changes in host DNA methylation may be associated with cervical cancer [15,19]. In 2016, the Taiwan Food and Drug Administration approved *PAX1* methylation as an adjunct to cytological testing for cervical cancer screening [20]. In a recent meta-analysis, Kelly et al. identified three genes (*CADM1*, *MAL*, and *PAX1*) as the most promising methylation markers for cervical cancer detection [19]. Previous studies have analyzed these genes using real-time PCR (qPCR); however, we propose that digital droplet PCR (ddPCR), a more advanced technique, may enhance their diagnostic potential. ddPCR offers exceptional resistance to PCR inhibitors and superior sensitivity for detecting low target concentrations [21]. Its endpoint counting strategy and reduced competition for reagents enable the accurate quantification of target nucleic acids even in the presence of inhibitors. Compared to qPCR, ddPCR demonstrates a lower detection limit and greater reliability in quantifying low-copy-number targets [22].

Therefore, this study aimed to determine the diagnostic potential of methylation analysis in the *CADM1*, *MAL*, and *PAX1* genes using ddPCR in women with HPV and CIN/CC.

## 2. Materials and Methods

### 2.1. General Information

The institution’s Local Ethics Committee approved this study (#01/22, 24 January 2022), and the researchers conducted the study in accordance with the Declaration of Helsinki. They enrolled patients between January 2022 and October 2023, obtaining signed informed consent from all participants. The study included 99 patients scheduled for cervical screening. The inclusion criterion of the experimental group was as follows: HPV-positive cases. The researchers assigned 73 patients to the experimental group. They recorded the following distribution of diagnostic results: 34 cases had no intraepithelial lesion or malignancy (NILM), 15 cases had low-grade squamous intraepithelial lesions (LSIL), 12 cases had high-grade squamous intraepithelial lesions (HSIL), and 12 cases had CC. The control group consisted of patients with negative cytology results and no evidence of HPV infection (n = 26). Patients with HSIL and cervical cancer underwent cervical biopsies via colposcopy-directed biopsy or conization for histologic diagnosis. The demographic and clinical characteristics of the study participants are summarized in Table 1, whereas more detailed data are available in Appendix A.

### 2.2. Preanalytical Biomaterial Processing

Cervical swabs from all participants in this study were collected, and the samples were placed in sealed vials containing a transport medium with a mucolytic agent and stored at −20 °C. DNA was extracted from the samples using the PROBA-NK-PLUS kit (DNA-Technology, Inc., Moscow, Russia). The vials containing purified DNA were then immediately frozen at −80 °C. Following thawing at room temperature, the samples were thoroughly mixed by pulse vortexing and centrifuged at 11,000× *g* for 1 min. None of the samples underwent more than one freeze–thaw cycle. The bisulfite conversion of the DNA was performed using the BisQuick kit (Evrogen, Inc., Moscow, Russia). To assess the quality of the extracted DNA, a spectrophotometric analysis of all samples was conducted using a NanoDrop 2000 device (Thermo Fisher Scientific, Inc., Waltham, MA, USA). The completeness of the bisulfite conversion of DNA was verified by a qPCR reaction using oligonucleotides designed to amplify a specific non-converted fragment of the *ACTB* gene (Appendix A, Appendix A).

### 2.3. HPV Detection and Genotyping

HPV detection and genotyping were performed using the HPV Quant-21 reagent kit and the DTprime amplifier (DNA-Technology, Inc., Moscow, Russia). According to the manufacturer’s specifications, the kit has an analytical sensitivity of 5 copies of the HPV genome per qPCR reaction. HPV strains were categorized as high-risk or low-risk based on a previous epidemiological investigation [15]. High-risk types include HPV 16, HPV 18, and other common types (26, 31, 33, 35, 39, 45, 51, 52, 53, 56, 58, 59, 66, 68, 73, and 82).

### 2.4. Methylation-Specific Polymerase Chain Reaction (MSP)

This study conducted methylation analysis of *PAX1*, *CADM1*, and *MAL* genes using both ddPCR and real-time PCR. The *ACTB* gene served as an internal control. The QX200 AutoDG ddPCR system (Bio-Rad Laboratories, Inc., Hercules, CA, USA) generated and read droplets for ddPCR analysis. The Veriti 96-Well Thermal Cycler (Life Technologies Corporation, Carlsbad, CA, USA) carried out the amplification. All ddPCR procedures were performed according to the manufacturer’s instructions. The thermocycling protocol for ddPCR included incubation at 95 °C (10 min), 45 cycles of denaturation at 94 °C (30 s) annealing/extension at 54 °C for *MAL/ACTB* and *CADM1/ACTB* and 60 °C for *PAX1/ACTB* (1 min), and incubation at 98 °C (10 min). The reaction mixture was prepared at a volume of 22 μL using 11 μL of a ready ddPCR Supermix for Probes (Bio-Rad Laboratories, Inc., Hercules, CA, USA), 0.9 μM primers, 0.25 μM probes, and 2 μL of bisulfite-converted DNA. For methylation analysis using real-time PCR, amplification was carried out on the CFX96 Touch System (Bio-Rad Laboratories, Inc., CA, USA) using the following protocol: incubation at 95 °C (5 min), 45 cycles of denaturation at 95 °C (15 s), and annealing/extension at 64 °C for *PAX1/ACTB*, 62 °C for *CADM1/ACTB*, and 60 °C for *MAL/ACTB* (30 s). The mixture was prepared at a volume of 20 μL using 4 μL of qPCRmix-HS (Evrogen, Inc., Moscow, Russia), 0.4 μM primers, 0.2 μM probes, and 2 μL of DNA. The Appendix A (Appendix A) present the sequences of oligonucleotide primers and probes used in the methylation-specific ddPCR and qPCR. The study team validated the analytical sensitivity and specificity of the assays through a series of experiments using Qiagen control methylated and unmethylated DNA (Qiagen GmbH, Hilden, Germany) mixed with methylated DNA fractions ranging from 0% to 100%. The qPCR assays detected methylation with limits of 10% for *PAX1* and 15% for *CADM1* and *MAL*, while the ddPCR assays achieved a detection limit below 1%. However, further dilution of methylated DNA prevented precise determination. The team analyzed samples in a single measurement for both ddPCR and qPCR. They repeated reactions for samples that did not meet quality criteria. The qPCR quality criteria included an *ACTB* (internal control) fluorescence threshold cycle of ≤32; the ddPCR quality criteria included the generation of >10,000 droplets per well and an *ACTB* level of >100 copies per well. To control batch variability across multiple qPCR and ddPCR runs, the team assessed reproducibility by ensuring that results from a positive control sample—containing a 1:1 mixture of Qiagen control methylated and unmethylated DNA (Qiagen GmbH, Hilden, Germany)—deviated by no more than 10%. False-positive signals across batches were determined (and subtracted from the results of the corresponding samples) based on the threshold cycle or the number of methylation-positive droplets detected in a sample containing Qiagen control unmethylated DNA (Qiagen GmbH, Hilden, Germany).

### 2.5. Statistical Analysis

The results of quantitative DNA analysis were presented in two forms: the total host DNA level (the level of the unmethylated portion of the *ACTB* gene in copies/μL of the reaction mixture) and DNA methylation (%). DNA methylation (%) measured by qPCR was calculated using the following formula: methDNAqPCR(%) = 2^(−ΔΔCt)^/(2^(−ΔΔCt)^ + 1) × 100, where “Ct” denotes the threshold fluorescence cycle. At the same time, ΔΔCt = ΔCtsample − ΔCtcontrol, where “control” is the sample containing 1 µL of Qiagen control methylated DNA and 1 µL of Qiagen control unmethylated DNA (Qiagen GmbH, Hilden, Germany). For ΔCtsample and ΔCtcontrol, ΔCt = Ctmeth − Ctunmeth. The values of DNA methylation (%) obtained by ddPCR were calculated using the following formula: methDNAddPCR(%) = [methylated target gene concentration (copies/μL)/level of the unmethylated portion of the *ACTB* gene (copies/μL)] × 100. The distribution of data was assessed for normality using the Shapiro–Wilk test. Given that none of the variables exhibited a normal distribution, nonparametric statistical tests were used. Paired quantitative and qualitative data were compared using the Wilcoxon signed-rank test and McNemar’s test, respectively. In contrast, unpaired quantitative and qualitative data underwent analysis using the Mann–Whitney U test and Fisher’s exact test, respectively. The results are presented as medians with interquartile ranges (Q1 and Q3) or percentiles 10 and 90 in cases with low detection rates for some of the studied variables. Receiver operating characteristic (ROC) analysis was used to evaluate the accuracy of binary classification models for specific variables. The Youden index was used to determine the optimal threshold values during the analysis of the ROC curve. No adjustments for multiple comparisons were performed, as no simultaneous comparisons of various groups were conducted. Spearman’s rank correlation coefficient (rs) was utilized to assess the relationship between two variables, with the magnitude of the correlation determined using Chaddock’s scale.

## 3. Results

A total of 91 specimens were deemed appropriate for methylation analysis among the 99 samples examined. Total host DNA levels, measured as the number of *ACTB* gene copies per µL of the reaction mixture, were compared across four patient groups: NILM, LSIL, HSIL, carcinoma, and control group. Median DNA levels were comparable among the groups, with no statistically significant differences observed (*p* > 0.05) (Figure 1). This suggests that the total DNA quantity in cervical smear samples does not vary significantly with cytological or histological severity. In the total cohort, the mean host DNA level was 165.99 copies/µL, while the maximum and minimum levels were 1009.93 copies/µL and 0.31 copies/µL, respectively.

### 3.1. Comparison of ddPCR and qPCR Performance

Detection rates of *CADM1, MAL*, and *PAX1* methylation were significantly higher when assessed by ddPCR, compared to qPCR. For *CADM1* (Figure 2a), the methylation detection rate using ddPCR reached 22.2%, while qPCR failed to detect methylation in any of the samples (0.0%; *p* = 0.001). For the *MAL* gene (Figure 2b), methylation was detected in 11.1% of the samples by ddPCR, whereas qPCR detected methylation in only 5.6% (*p* = 0.001). For *PAX1* (Figure 2c), ddPCR revealed methylation in 77.8% of the cases, in contrast to 0.0% detected using qPCR (*p* = 0.001).

ddPCR detected significantly higher methylation levels than qPCR for all analyzed genes (*p* < 0.05). Specifically, ddPCR revealed marked elevations in the methylation levels of *CADM1* (Figure 3a), *MAL* (Figure 3b), and *PAX1* (Figure 3c). For each gene, the distribution of methylation values shifted significantly toward higher levels with ddPCR compared to with qPCR. Statistical analysis using McNemar’s test confirmed that these differences were significant: *p* = 0.001 for *CADM1* and *PAX1*, and *p* = 0.008 for *MAL*.

These results demonstrate that ddPCR exhibited significantly higher analytical performance than qPCR for all three targets. Additionally, qPCR showed significantly lower discriminative power in the experimental group (diagnostic performance is shown in Appendix A). Consequently, this study shifted its focus to analyzing the results obtained using ddPCR.

Figure 4 shows an example of ddPCR-derived 2D fluorescence plots for positive and negative samples for the *CADM1* gene methylation analysis. Additional examples of 2D plots for other analyzed genes are provided in Appendix A.

### 3.2. Comparison of CADM1, MAL, and PAX1 Methylation Between HPV-Positive and HPV-Negative Patients

The detection rate of *CADM1* methylation was significantly higher in the high-risk HPV-positive group compared to the HPV-negative group (30.7% vs. 3.7%, respectively; *p* = 0.005; Fisher’s exact test) (Figure 5a). Consistently with this finding, the quantitative *CADM1* methylation level was also significantly elevated in the HPV-positive group (0 [0; 3.01]%) compared to HPV-negative samples (0 [0; 0]%, respectively; *p* = 0.005; Figure 5b).

Compared to the HPV-negative group, the HPV-positive group demonstrated higher median methylation levels for both the *MAL* (0 [0; 0.03]% vs. 0 [0; 0.43]%, respectively; *p* = 0.593) and *PAX1* (1.32 [0; 10.81]% vs. 0.4 [0.01; 3.18]%; *p* = 0.536) genes. However, these differences were not statistically significant (Figure 5d,f). The detection rates of *MAL* (24.6% vs. 17.9%; *p* = 0.346) and *PAX1* (69.6% vs. 78.6%; *p* = 0.051) methylation also did not differ significantly between HPV-positive and HPV-negative groups (Figure 5c,e).

### 3.3. Evaluation of the Diagnostic Potential of CADM1, MAL, and PAX1 Methylation

Significant differences in methylation levels of *CADM1*, *MAL*, and *PAX1* were observed across diagnostic categories using the Kruskal–Wallis test (*p* < 0.001 for *CADM1* and *MAL*; *p* = 0.010 for *PAX1*) (Table 2). Pairwise comparisons were subsequently performed using the Mann–Whitney U test. For *CADM1*, methylation levels in the carcinoma group were significantly higher than those in NILM (*p* = 0.042), LSIL (*p* = 0.01), and HSIL (*p* < 0.001). No significant differences were observed among NILM, LSIL, and HSIL. For *MAL*, a significant increase in methylation was found only in the carcinoma group compared to all other categories (vs. NILM: *p* = 0.049; vs. LSIL: *p* = 0.006; vs. HSIL: *p* = 0.010), while the NILM, LSIL, and HSIL groups did not differ significantly. Regarding *PAX1*, methylation levels were significantly higher in carcinoma than in NILM (*p* = 0.003), LSIL (*p* = 0.003), and HSIL (*p* = 0.015), while comparisons among the non-carcinoma groups did not yield statistically significant differences. These findings suggest that significant epigenetic changes in *CADM1, MAL*, and *PAX1* are primarily associated with malignant transformation rather than a gradual increase in methylation across pre-invasive stages.

Multivariate logistic regression analysis demonstrated that *CADM1* methylation was an independent predictor of high-grade cervical lesions (HSIL+) after adjusting for age, smoking, parity, and the number of sexual partners (OR = 39.546; 95% CI: 1.130–1383.704; *p* = 0.043). However, *MAL* (OR = 1.981; *p* = 0.632) and *PAX1* (OR = 1.557; *p* = 0.479) methylation did not reach statistical significance in the model.

Correlation analysis showed no strong associations between methylation levels and clinical–demographic factors (age, parity, smoking, number of sexual partners). Appendix A presents the correlation matrix.

To evaluate the diagnostic relevance of these methylation changes, ROC curve analyses were conducted for the *CADM1, MAL*, and *PAX1* genes. First, we assessed their ability to distinguish HPV-negative NILM cases from histologically confirmed LSIL (Figure 6a), HSIL (Figure 6b), and carcinoma (Figure 6c). Among these comparisons, *PAX1* methylation demonstrated statistically significant diagnostic utility in differentiating carcinoma from NILM, with increased sensitivity and specificity. In contrast, *CADM1* and *MAL* showed no significant diagnostic value for carcinoma detection in this context.

Further analysis of marker performance in distinguishing high-grade (HSIL+) from non-high-grade abnormalities (NILM/LSIL) and abnormal cytology (LSIL+) from normal cytology (NILM) is presented in Figure 7a,b. Among the individual genes, *CADM1* methylation showed the highest diagnostic accuracy for detecting HSIL+ (AUC = 0.671), followed by *MAL* (AUC = 0.643) and *PAX1* (AUC = 0.610) (Figure 6d). Similarly, *CADM1* was the most accurate in distinguishing abnormal from normal cytology (AUC = 0.665), while *MAL* and *PAX1* demonstrated lower AUCs of 0.524 and 0.430, respectively (Figure 6e).

The combination of *CADM1* and *MAL* methylation enhanced the overall diagnostic performance for HSIL+ detection, as evidenced by a higher AUC (Figure 7a). Although this combination slightly reduced sensitivity, it improved specificity compared to individual markers. Full results are presented in Appendix A.

The best diagnostic performance for carcinoma was achieved with the combined methylation analysis of *CADM1* and *MAL* (AUC = 0.912), which outperformed the individual markers: *CADM1* (AUC = 0.770), *MAL* (AUC = 0.770), and *PAX1* (AUC = 0.813) (Figure 7b). At the optimal cut-off values (4.57% for *CADM1*, 0.085% for *MAL*, 1.745% for *PAX1*, and 2.785% for *CADM1+MAL*), the corresponding sensitivities were 50.0%, 62.5%, 87.5%, and 70.0%, whereas the specificities were 100.0%, 96.9%, 68.7%, and 100.0%, respectively. Details are provided in Appendix A.

## 4. Discussion

*PAX1*, *MAL*, and *CADM1* have been proposed as tumour suppressor genes involved in the development and progression of cervical cancer [23,24,25,26]. The aberrant methylation of their promoter regions is a well-documented mechanism leading to transcriptional silencing and has been associated with HPV-induced neoplastic transformation [15,27]. In this study, we assessed the diagnostic performance of the methylation-based detection of these genes using ddPCR, which provides enhanced sensitivity and quantification compared to conventional qPCR.

Our results demonstrated a significant association between gene methylation levels and the histological severity of cervical lesions. *CADM1* methylation exhibited a stepwise increase across the NILM, LSIL, HSIL, and carcinoma groups, with a statistically significant elevation in cancer cases (*p* < 0.001). These findings align with earlier studies that reported dense *CADM1* promoter methylation and loss of protein expression in high-grade lesions and cervical carcinomas [28,29]. *MAL* methylation was detected exclusively in carcinoma samples, supporting its potential utility as a late-stage marker [27]. In contrast, *PAX1* methylation was already present in a subset of NILM and LSIL cases. It showed a marked increase in carcinoma, indicating that it may emerge earlier in cervical carcinogenesis, although with limited specificity in non-malignant lesions [24,30].

These observations support the hypothesis that *CADM1* and *MAL* methylation are relatively late events in the progression of cervical disease, while *PAX1* methylation may occur in an earlier stage. This highlights the potential benefit of multiplex marker panels combining early and late events to improve sensitivity and specificity across the full spectrum of cervical lesions [31].

ROC curve analyses further confirmed the superior performance of the combined *CADM1/MAL* methylation panel in detecting carcinoma (AUC = 0.912), compared to *CADM1* and *MAL* alone (both AUC = 0.770) and *PAX1* (AUC = 0.813). The panel also achieved 100% specificity and 70% sensitivity at optimal cut-off thresholds. These findings are consistent with previous studies that support the use of *CADM1/MAL* as an effective triage tool in HPV-positive women [21,31].

Despite the strong performance, individual markers showed sensitivity levels below 80%, which limits their potential as primary screening tools. However, their value in triage—especially when used alongside HPV testing—remains significant. DNA methylation testing can enhance the specificity of colposcopy referrals in women with HPV-positive cytology and atypical findings [21]. This is particularly relevant in low-resource settings, where overtreatment remains a concern.

A crucial methodological advantage of our study was the use of ddPCR, which significantly outperformed qPCR in methylation detection. ddPCR identified *CADM1* and *PAX1* methylation in 22.2% and 77.8% of cases, respectively, while qPCR detected none. These results confirm the superior sensitivity and reproducibility of ddPCR, especially for detecting rare methylated alleles or low-copy targets in clinical specimens [21,32,33]. ddPCR has been widely validated in oncology for the detection of somatic mutations and DNA methylation profiling due to its low susceptibility to PCR inhibitors and its ability to perform precise endpoint quantification [34]. However, ddPCR is not without limitations. The high costs of equipment and reagents, as well as limited access to ddPCR platforms in many clinical laboratories, currently hinder its widespread adoption.

In contrast, qPCR remains more accessible and economical for routine diagnostics [33]. These practical barriers must be addressed before ddPCR can be routinely implemented in cervical cancer screening programmes. Recent studies have also highlighted the potential of methylation markers for triaging HPV-positive women with minor cytological abnormalities, such as ASC-H and CIN1 [35].

Moreover, the limited number of HSIL and carcinoma cases reflects the exploratory nature of this pilot study. Our primary aim was not to perform a fully powered clinical trial but rather to establish a methodological basis for future research by comparing two analytical approaches—qPCR and ddPCR—for methylation detection in HPV-positive women.

Additionally, both ddPCR and qPCR rely on the bisulfite conversion of DNA, a process that may introduce methodological biases. Incomplete conversion, DNA degradation, or loss during processing may affect the accuracy of downstream methylation analysis. These challenges are especially pronounced in low-input or degraded clinical samples, emphasizing the need for robust conversion protocols and internal controls [31,35].

Our findings are consistent with global data that link elevated DNA methylation in HPV-infected cervical cells to increasing CIN grade and cancer risk [15,27]. A recent Russian review also emphasized the clinical relevance of methylation markers and the importance of bisulfite optimization for reliable detection in diagnostics [36]. The integration of highly sensitive platforms like ddPCR and validated gene panels such as *CADM1/MAL* represents a promising direction in the development of molecular strategies for cervical cancer triage, particularly among HPV-positive women.

This study has several limitations that should be taken into account when interpreting the results. First, the relatively small sample size limits the generalizability of our findings, particularly for subgroup analyses. Second, the study was conducted at a single centre, which may introduce selection bias. Future multicenter studies involving larger and more diverse populations, as well as longitudinal follow-up, are warranted to confirm these findings and support their translation into clinical practice. Due to the limited sample size, internal validation (e.g., cross-validation) was not implemented. The cutoff values for the presented assays require further verification in a larger cohort.

## 5. Conclusions

This study demonstrated that the promoter methylation of the *CADM1*, *MAL*, and *PAX1* genes is significantly associated with the severity of cervical epithelial abnormalities in HPV-positive women. Among the tested biomarkers, the combined methylation panel of *CADM1/MAL* exhibited the highest diagnostic performance for distinguishing histologic HSIL+ and carcinoma from low-grade lesions and NILM. *MAL* methylation was highly specific for carcinoma, whereas *PAX1* methylation appeared earlier in the disease course, suggesting its potential utility for detecting precancerous changes.

Although methylation-based assays demonstrated high specificity, their moderate sensitivity supports their use primarily as a triage tool in combination with HPV testing rather than as a standalone screening method. Moreover, ddPCR provided significantly improved methylation detection compared to real-time PCR, highlighting its value as a precise and sensitive platform for epigenetic diagnostics. However, the broader clinical implementation of ddPCR may be limited by its higher cost and restricted availability in routine laboratory settings. Additionally, the requirement for bisulfite conversion in methylation assays can introduce DNA degradation or incomplete conversion, which may affect quantification accuracy.

Future multicenter studies involving larger and more diverse populations, along with longitudinal follow-up, are warranted to validate these findings and further assess the clinical applicability of DNA methylation analysis, particularly using ddPCR as a non-invasive method for cervical cancer screening and risk stratification.

## Figures and Tables

**Figure 1 biomedicines-13-01450-f001:**
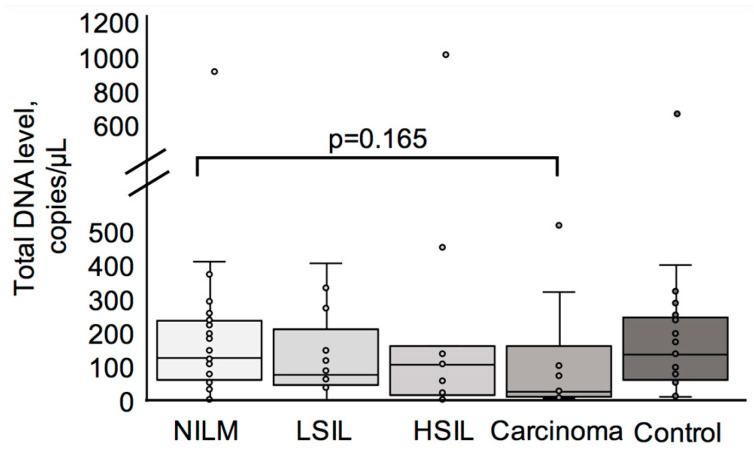
A comparison of total DNA level analysis results in smear samples of patients in different groups. Data are presented as copies per 1 μL of the reaction mixture. Patients in the NILM, LSIL, HSIL, and carcinoma groups were HPV-positive, and those in the control group were HPV-negative. Total DNA level was assessed based on the level of the unmethylated portion of the *ACTB* gene by ddPCR. The *p*-value was obtained using the Kruskal–Wallis test.

**Figure 2 biomedicines-13-01450-f002:**
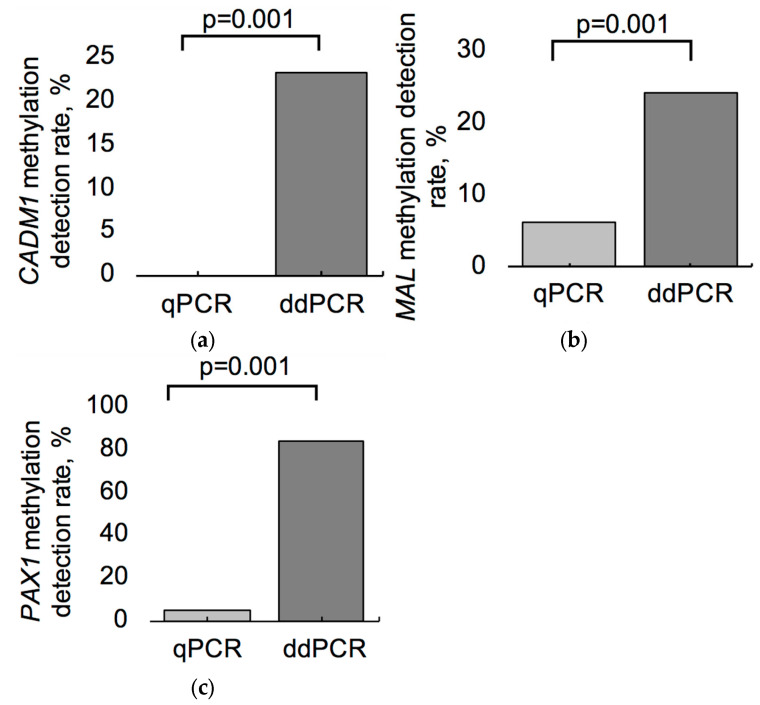
Comparison of methylation detection rates for *CADM1* (**a**), *MAL* (**b**), and *PAX1* (**c**) genes in smear samples using qPCR and ddPCR methods. All data are presented as methylation detection rates (%) in total cohort. *p*-values were obtained using McNemar’s test.

**Figure 3 biomedicines-13-01450-f003:**
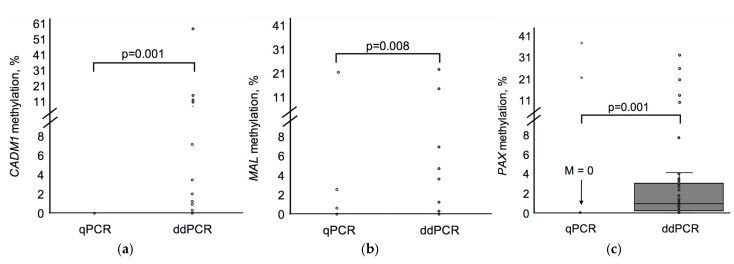
A comparison of gene methylation levels assessed using qPCR and ddPCR methods: (**a**) *CADM1* methylation; (**b**) *MAL* methylation; (**c**) *PAX1* methylation. Data are presented as individual values with summary statistics (median and interquartile range) in the total patient cohort (%). *p*-values were obtained using the Wilcoxon test. In (**a**,**b**), and in (**c**) for the qPCR data, boxplots are not visible due to low or zero median methylation levels.

**Figure 4 biomedicines-13-01450-f004:**
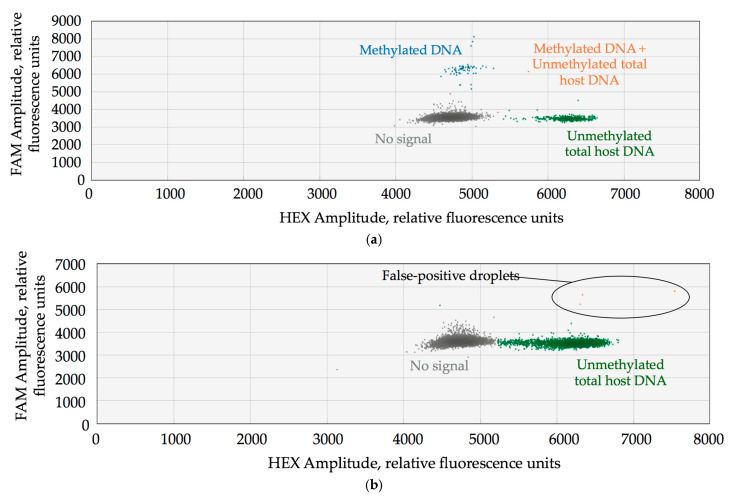
Representative 2D fluorescence amplitude plots of ddPCR results for the *CADM1* gene. Probes specific for methylated *CADM1* DNA are labelled with FAM, whereas probes for unmethylated total host DNA (*ACTB*) are labelled with HEX. (**a**) A positive sample showing clusters corresponding to methylated *CADM1* DNA (FAM), unmethylated ACTB DNA (HEX), and double-positive droplets (FAM + HEX). (**b**) A negative sample showing only unmethylated *ACTB DNA*, with a few apparent false-positive droplets detected in the upper right quadrant.

**Figure 5 biomedicines-13-01450-f005:**
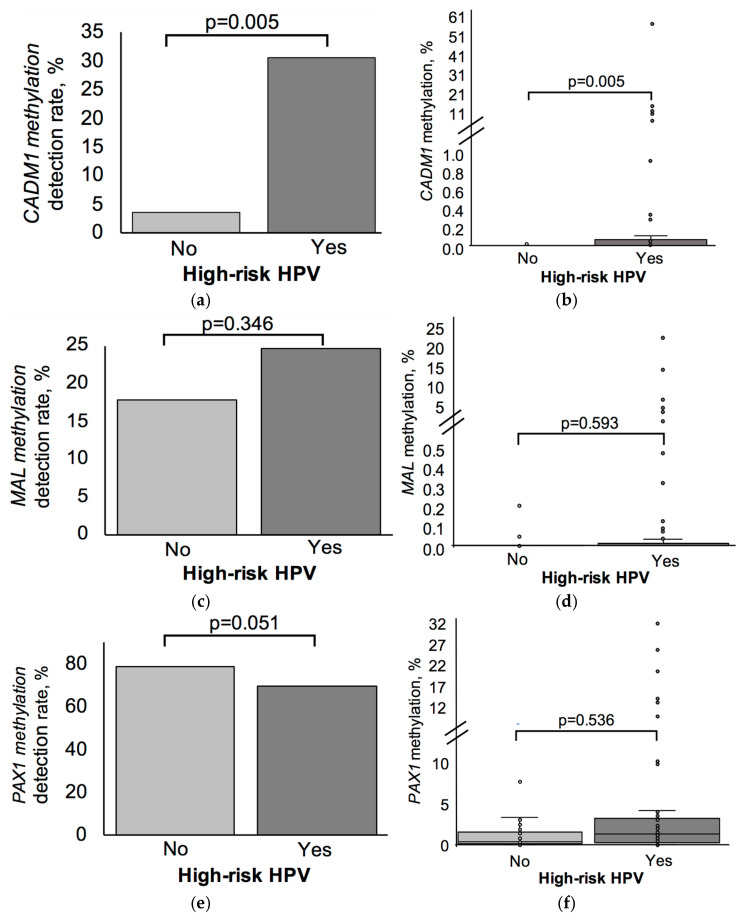
Association between high-risk HPV status and methylation of *CADM1*, *MAL*, and *PAX1*. (**a**,**c**,**e**) Methylation detection rates (%) based on ddPCR results. (**b**,**d**,**f**) Quantitative methylation levels (%) are presented as box plots. Analysis was performed on the total patient cohort using ddPCR. High-risk HPV types included HPV 16, 18, 26, 31, 33, 35, 39, 45, 51, 52, 53, 56, 58, 59, 66, 68, 73, and 82. *p*-values were obtained using Fisher’s exact test (for detection rates) and the Mann–Whitney U test (for methylation levels).

**Figure 6 biomedicines-13-01450-f006:**
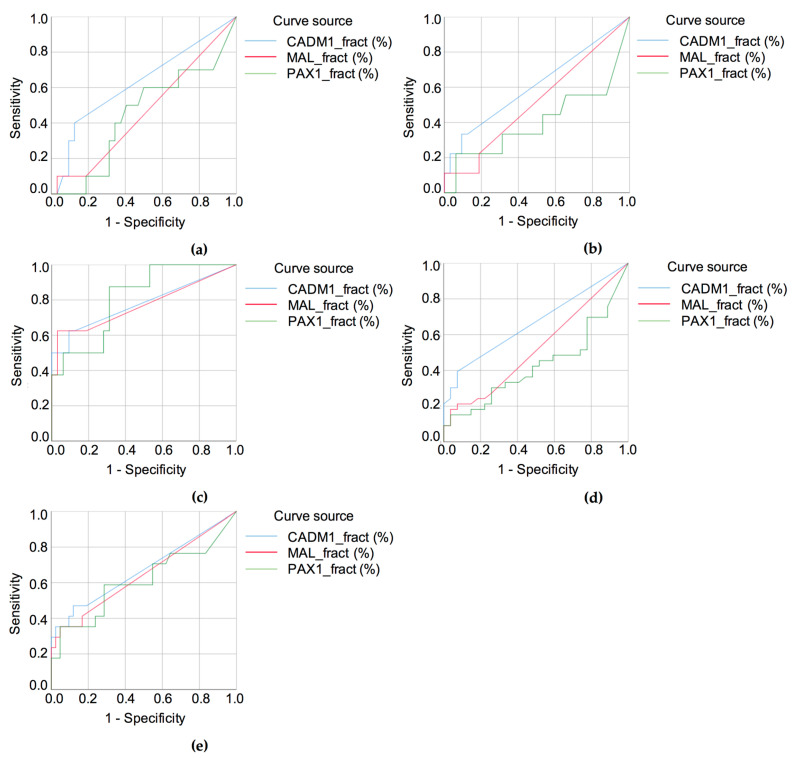
ROC curve analysis based on *CADM1*, *MAL*, and *PAX1* methylation levels for distinguishing HPV-positive NILM from various lesion categories. (**a**) NILM (HPV+) vs. LSIL. (**b**) NILM (HPV+) vs. HSIL. (**c**) NILM (HPV+) vs. carcinoma. (**d**) NILM (HPV+) vs. LSIL + HSIL + carcinoma. (**e**) NILM (HPV+) + LSIL vs. HSIL + carcinoma.

**Figure 7 biomedicines-13-01450-f007:**
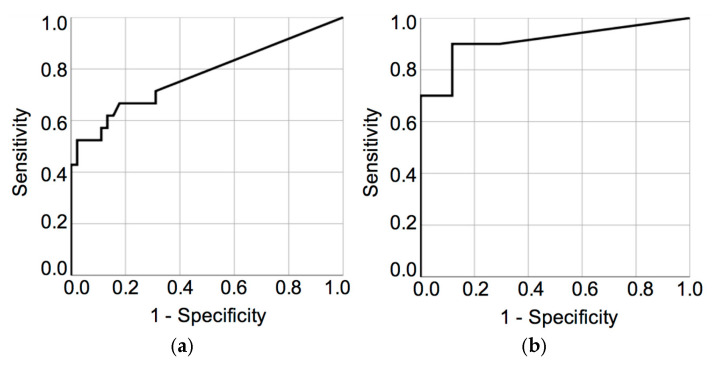
ROC curve analysis based on combined *CADM1* and *MAL* methylation markers. (**a**) NILM (HPV+) + LSIL vs. HSIL + carcinoma. (**b**) NILM (HPV+) vs. carcinoma.

**Table 1 biomedicines-13-01450-t001:** The demographic and clinical characteristics of the study participants.

Parameters	NILM (n = 34)	LSIL (n = 15)	HSIL (n = 12)	Carcinoma (n = 12)	Control Group (n = 26)
Age, years ^1^	29 (18:40)	25 (21:32)	35 (21:49)	44 (30:70)	27 (21:43)
HPV status:					
16/18, *n*	20 (58.8%)	11 (73.3%)	9 (75%)	11 (91.7%)	0 (0%)
other HPV, *n*	28 (82.4%)	13 (86.7%)	8 (66.7%)	5 (41.7%)	0 (0%)
Smoking:					
yes, *n*	4 (11.8%)	7 (46.7%)	4 (33.3%)	4 (33.3%)	3 (11.5%)
no, *n*	30 (88.2%)	8 (53.3%)	8 (66.7%)	8 (66.7%)	23 (88.5%)
Number of pregnancies:					
<3, *n*	33 (97.1%)	15 (100%)	10 (83.3%)	9 (75%)	25 (96.2%)
≥3, *n*	1 (2.9%)	0 (0%)	2 (16.7%)	3 (25%)	1 (3.8%)
Condom use:					
yes, *n*	14 (41.2%)	8 (53.3%)	1 (8.3%)	1 (8.3%)	19 (73.1%)
no, *n*	20 (58.8%)	7 (46.7%)	11 (91.7%)	11 (91.7%)	7 (26.9%)
Number of sexual partners:					
<4, *n*	30 (88.2%)	12 (80%)	8 (66.7%)	7 (58.3%)	22 (84.6%)
≥4, *n*	4 (11.8%)	3 (20%)	4 (33.3%)	5 (41.7%)	4 (15.4%)
Years of sexual activity:					
<10, *n*	16 (47.1%)	13 (86.7%)	3 (25%)	0 (0%)	17 (65.4%)
≥10, *n*	18 (52.9%)	2 (13.3%)	9 (75%)	12 (100%)	9 (34.6%)

NILM—negative test for intraepithelial lesion or malignancy; LSIL—low-grade squamous intraepithelial lesion; HSIL—high-grade squamous intraepithelial lesion; HPV—human papillomavirus; ^1^ data presented as mean (range); values presented as absolute numbers and relative proportions (%).

**Table 2 biomedicines-13-01450-t002:** Methylation levels of *CADM1*, *MAL*, and *PAX1* genes across cervical diagnostic categories (NILM, LSIL, HSIL, carcinoma).

Pap Test/Biopsy	*CADM1* (%)	*MAL* (%)	*PAX1* (%)
NILM	0 [0; 0]	0 [0; 0]	0.77 [0.22; 2.75]
LSIL	0 [0; 0.04]	0 [0; 0]	0.91 [0.03; 1.53]
HSIL	0 [0; 0.63]	0 [0; 0]	0.63 [0.1; 1.73]
Carcinoma	2.37 [0.07; 9.61]	1.97 [0.02; 6.37]	6.49 [2.23; 22.02]
*p*-value *	<0.001	<0.001	0.010

* *p*-values are from Kruskal–Wallis tests; data presented as median [quartile 1; quartile 3].

## Data Availability

The data presented in this study are available in Appendix A.

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
