# Peer review of "Detection of CADM1, MAL, and PAX1 Methylation by ddPCR for Triage of HPV-Positive Cervical Lesions"

_biomedicines, 2025, doi:10.3390/biomedicines13061450_

Round 1
Reviewer 1 Report
Comments and Suggestions for Authors This manuscript by Anisimova et al., evaluates the diagnostic performance of CADM1, MAL, and PAX1 promoter methylation, measured using droplet digital PCR (ddPCR), in detecting high-grade cervical lesions and carcinoma in HPV-positive women. The study presents ddPCR as a superior method to qPCR in terms of sensitivity and specificity, with a particular emphasis on the CADM1/MAL methylation panel.Sample Size and Power:
While the results are promising, the relatively small sample size, particularly in the carcinoma (n = 12) and HSIL (n = 12) groups, limits both statistical power and generalizability. How did the authors calculate power?
Interpretation of Results:
The discussion appropriately contextualizes the findings within the existing literature. However, further commentary on the clinical implications, such as cost-benefit trade-offs of ddPCR implementation in screening programs, would strengthen the impact. In addition, I suggest including the following reference because currently, Double Methylation as Screening for ASC-H- and CIN1 HPV-Positive Women is used (https://doi.org/10.3390/pathogens13040312)
Author Response
Dated: 25.05.2025.
Subject: Submission of revised manuscript
To the Editor of the BIOMEDICINES journal.
Thank you for giving us the opportunity to revise our manuscript titled “The Significance of Genomic DNA Methylation in Identifying Cytological and Histological Abnormalities of the Cervix in Women with High-Risk Human Papillomavirus Infection” (Manuscript ID: biomedicines-3649339). We are grateful for the time and effort dedicated to reviewing our work and for the constructive comments provided. We have carefully revised the manuscript and addressed all suggestions. The changes are highlighted in the revised version using the track changes function in MS Word.
Below is a point-by-point response to the reviewer’s comments.
Comments from Reviewer 1
Comment 1:
“Sample Size and Power:
While the results are promising, the relatively small sample size, particularly in the carcinoma (n = 12) and HSIL (n = 12) groups, limits both statistical power and generalizability. How did the authors calculate power?”
Response:
We thank the reviewer for this important comment. We agree that the small number of cases in the HSIL and carcinoma groups represents a limitation. This study was designed as a pilot investigation, primarily aimed at establishing a methodological basis for comparing ddPCR and qPCR in detecting gene methylation. Given its exploratory nature, the overall sample size was not based on a formal power calculation for the full cohort. However, for the specific objective of evaluating the discriminative potential of methylation biomarkers via ROC curve analysis in the small groups noted by the reviewer, we performed a sample size estimation. Assuming an expected AUC of at least 0.7, a Type I error of 0.05, a Type II error of 0.20 (i.e., 80% power), and a negative-to-positive sample ratio of at least 2:1, the required number of patients in the positive group was calculated to be 19. In our study, the combined HSIL and carcinoma group exceeded this threshold, allowing sufficient power for pooled ROC analysis. However, we acknowledge that the statistical power is lower when these subgroups are considered separately. The sample size estimation was based on the method described by Hanley and McNeil (Radiology, 1982; DOI: 10.1148/radiology.143.1.7063747).
The following clarification has been added to the Discussion (Page 11, lines 384–388):
“Moreover, the limited number of HSIL and carcinoma cases reflects the exploratory nature of this pilot study. Our primary aim was not to perform a fully powered clinical trial, but rather to establish a methodological basis for future research by comparing two analytical approaches—qPCR and ddPCR—for methylation detection in HPV-positive women.”
Comment 2:
“Interpretation of Results:
The discussion appropriately contextualizes the findings within the existing literature. However, further commentary on the clinical implications, such as cost-benefit trade-offs of ddPCR implementation in screening programs, would strengthen the impact. In addition, I suggest including the following reference because currently, Double Methylation as Screening for ASC-H- and CIN1 HPV-Positive Women is used (https://doi.org/10.3390/pathogens13040312)”
We thank the reviewer for this valuable suggestion. The limitations of ddPCR implementation, including high equipment costs, the need for specialized infrastructure, and the requirement for trained personnel, have already been discussed in the Discussion section of the manuscript. Specifically, this is reflected in the sentence: “However, ddPCR is not without limitations. High costs of equipment and reagents, as well as limited access to ddPCR platforms in many clinical laboratories, currently hinder its widespread adoption.”
We also thank the reviewer for suggesting the relevant reference by Peronace et al. (2024). We have cited this work in the revised Discussion to support the role of methylation markers in triaging HPV-positive women with minor cytological abnormalities. The following sentence has been added (Page 11, lines 381-383): “Recent studies have also highlighted the potential of methylation markers for triaging HPV-positive women with minor cytological abnormalities, such as ASC-H and CIN1 [32].”
The reference has been added to the References list accordingly.
Reviewer 2 Report
Comments and Suggestions for Authors
The study highlights that promoter methylation of the CADM1, MAL, and PAX1 genes is significantly correlated with the severity of cervical epithelial abnormalities in HPV-positive women. The combined CADM1/MAL methylation panel demonstrated the highest diagnostic accuracy in distinguishing high-grade squamous intraepithelial lesions (HSIL+) and carcinoma from low-grade lesions and NILM. Notably, MAL methylation showed strong specificity for carcinoma, while PAX1 methylation was detectable earlier in the disease process, indicating its potential utility for identifying precancerous changes. Droplet digital PCR (ddPCR) significantly enhanced methylation detection sensitivity compared to real-time PCR, affirming its potential as a precise epigenetic diagnostic platform.
1. Any correlation with other factors like ZNF582, FAM19A4
2. The current statistical methods are solid butI would like to suggest that the use of Multivariable logistic regression analysis can help to determine whether methylation markers predict outcomes independently of other clinical variables.
3. Use of Cluster analysis or heatmaps may reveal subgroups of patients with distinct methylation signatures relevant for personalised risk stratification.
4. The figures can be enhanced in quality as some content are difficult to read figure 2, 6, and 7
5. Add recent literature
Author Response
Dated: 25.05.2025.
Subject: Submission of revised manuscript
To the Editor of the BIOMEDICINES journal.
Thank you for giving us the opportunity to revise our manuscript titled “The Significance of Genomic DNA Methylation in Identifying Cytological and Histological Abnormalities of the Cervix in Women with High-Risk Human Papillomavirus Infection” (Manuscript ID: biomedicines-3649339). We are grateful for the time and effort dedicated to reviewing our work and for the constructive comments provided. We have carefully revised the manuscript and addressed all suggestions. The changes are highlighted in the revised version using the track changes function in MS Word.
Below is a point-by-point response to the reviewer’s comments.
Comments from Reviewer 2
Comment 1:
“Any correlation with other factors like ZNF582, FAM19A4”
Response:
Thank you for this insightful comment. In the present study, we focused specifically on the methylation status of CADM1, MAL, and PAX1, which are the most established and commonly used markers in our region. Other markers such as ZNF582 and FAM19A4 were not included in our panel, as they are not currently part of routine clinical practice in our setting. Nonetheless, we acknowledge their relevance and will consider including them in future research.
Comment 2:
“The current statistical methods are solid butI would like to suggest that the use of Multivariable logistic regression analysis can help to determine whether methylation markers predict outcomes independently of other clinical variables.”
Response:
Thank you for this valuable suggestion. In our pilot study, we prioritized an analytical comparison of ddPCR versus qPCR for methylation detection. Given the small and diagnostically heterogeneous cohort—where some patients were triaged by cytology only and others by histology—a multivariable logistic model would be underpowered and potentially unstable. Importantly, several larger studies have demonstrated that promoter methylation of our markers is independent of standard clinical covariates: Meršaková et al. ( https://doi.org/10.3892/ol.2018.9505) applied logistic regression to evaluate CADM1 and MAL promoter methylation in 202 cervical cytology specimens and found that methylation levels stratified cases and controls independently of patient age and HPV genotype.
In a cohort of 188 women, Chen et al. (https://oss.ejgo.net/files/article/20240614-1257/pdf/EJGO2022092101.pdf?utm_source=chatgpt.com ) constructed a multivariate logistic regression model for PAX1 promoter methylation in exfoliated cervical cells and reported no significant associations between methylation levels and age, weight, height, age at menarche, or parity. These data confirm that clinical characteristics do not materially influence CADM1, MAL, or PAX1 methylation status, supporting our focus on assay performance. We will plan multivariable analyses in future, adequately powered studies with more uniform clinical datasets.
Comment 3:
“Use of cluster analysis or heatmaps may reveal subgroups of patients with distinct methylation signatures relevant for personalised risk stratification.”
Response:
We thank the reviewer for this interesting and relevant suggestion. However, due to the pilot character of our study and the limited size of individual subgroups, advanced statistical approaches such as cluster analysis and heatmap generation were not applied at this stage. Our primary aim was to evaluate the technical and diagnostic potential of ddPCR for methylation detection, in direct comparison to conventional real-time PCR.
It is important to note that the majority of patients included in this study were recruited through cervical cancer screening programs. This reflects our broader objective of improving the clinical accuracy of molecular diagnostic methods in real-world screening settings. Accordingly, the cohort predominantly consisted of HPV-positive women with NILM or LSIL cytology, while the number of HSIL cases was relatively small. The cervical cancer group, which represented the highest-grade pathology, was recruited separately from the Department of Oncogynecology at our university clinic and was not part of the screening population.
Initial exploratory attempts to identify methylation-based subgroups using unsupervised methods were inconclusive, likely due to the small sample size and imbalanced group structure. Nevertheless, based on the promising results obtained, we plan to expand both the sample size and the methylation panel in future phases of the project, and we intend to incorporate clustering techniques as part of the extended statistical analysis.
No changes were made to the manuscript in response to this comment due to the exploratory nature and limitations of the current dataset.
Comment 4:
“The figures can be enhanced in quality as some content are difficult to read (figure 2, 6, and 7).”
Response:
We thank the reviewer for this helpful comment. We have improved the resolution and clarity of Figures 2, 6, and 7 as suggested. The revised figures have been provided as a separate file and uploaded to the submission system under the “Graphical Images” section.
Comment 5:
“Add recent literature.”
Response:
We thank the reviewer for this valuable suggestion. We have added four recent publications to the References list ([1,2,16,34]) to strengthen the manuscript with up-to-date sources. As a result, 11 of the cited articles are from the past five years. The remaining references were deliberately retained, as they represent foundational studies that shaped the current understanding of HPV biology and DNA methylation, providing essential context for the present work.
Reviewer 3 Report
Comments and Suggestions for Authors
This manuscript examines the diagnostic value of DNA methylation markers (CADM1, MAL, and PAX1) in HPV-positive women with cervical lesions, emphasising the role of droplet digital PCR (ddPCR) for better detection. Integrating host gene methylation profiling with ddPCR enhances cervical cancer triage methods. The study is well-organised, with ddPCR providing methodological rigour. Results show the CADM1/MAL panel effectively distinguishes carcinoma cases. However, further clarifications could enhance the manuscript's scientific impact.
Key considerations include:-
Clarifying the study’s novelty in the title and introduction.
Adding details about sample handling, assay validation, and statistical methods.
Addressing confounding factors such as age differences and lack of matched controls.
Explaining the biological roles of the genes and their methylation patterns at various stages.
Discussing limitations regarding sensitivity for HSIL and comparing findings to current screening methods.
Standardising visual presentation and adding statistical annotations will improve data visualisation. In conclusion, this study shows promise with a revised context for its findings.
Title & Abstract
- While the study explores a well-established biomarker class (DNA methylation), its application of ddPCR adds technical novelty. However, this innovation is not evident in the title. I suggest revising the title to explicitly highlight both the use of ddPCR and the biomarkers evaluated to reflect the unique contribution of the work better.
- The abstract overviews the main findings and emphasises the CADM1/MAL panel’s diagnostic value using ddPCR. However, it would benefit from mentioning critical aspects such as the sample size, study design, and methodological limitations to better frame the context and avoid overestimating diagnostic accuracy.
Introduction
- The introduction offers a solid overview but lacks focus and urgency. It should more clearly articulate the gap in current screening strategies and position host gene methylation, mainly assessed via ddPCR, as a solution.
- Lines 36–40: The description of HPV’s genome and classification could be streamlined to focus more directly on diagnostic gaps in HPV screening practices.
- While accurate, the virological classification and L1 protein-cell interactions seem peripheral. Consider condensing this to highlight the clinical relevance of HPV in cervical carcinogenesis.
- The transition from HPV infection mechanisms to methylation is abrupt. Consider directly linking oncogenic HPV types to host gene methylation and neoplastic progression.
- Lines 50–56 provide essential context on the transient nature of HPV infections. However, I suggest linking this point more directly to the challenge of differentiating persistent, high-risk infections from benign ones.
- Lines 57–64 conflate host and viral methylation. Please clarify whether the current study focuses exclusively on host gene methylation and why this offers greater diagnostic utility.
- The sentence on CpG methylation in the HPV genome doesn’t align with the study’s host-focused methylation approach. Reorganising this to focus on host gene methylation would improve coherence.
- The mention of CADM1, MAL, and PAX1 as promising markers appears too late. Introducing them earlier would help build a stronger rationale for the study.
- Lines 76–83: While the benefits of ddPCR are mentioned, the rationale would be more compelling if you referenced known limitations of qPCR in detecting low-abundance methylation in cervical samples.
- The study's aim is broadly stated. Clarifying it in terms of diagnostic performance and comparison to qPCR would strengthen the focus.
- Consider explicitly stating a hypothesis: “We hypothesise that ddPCR enables more accurate detection of CADM1, MAL, and PAX1 methylation, thereby improving diagnostic discrimination between benign and malignant cervical lesions.”
- Overall, I recommend reducing excessive virological detail (e.g., lines 41–45) and emphasising the clinical limitations of current screening tools, the diagnostic role of host methylation, and the methodological advancement of ddPCR.
Methods
- The study design is generally well-explained, but the inclusion/exclusion criteria require a more precise definition.
- Were participants with prior cervical cancer or treatment history excluded?
- For NILM cases, please clarify if all were HPV-positive or if this group was mixed.
- Was a statistical power calculation performed to determine the adequacy of the sample size?
- How was participant selection or allocation handled to reduce potential sampling bias?
- Lines 95–97: The control group is acceptable, but not age-matched. Please clarify whether adjustments for confounders such as age, sexual history, and parity were applied.
- Was histopathology reviewed by multiple observers or in a blinded fashion relative to methylation results?
- Sample collection and handling are described, but pre-analytical variables (e.g., freezing time, thawing) are not addressed. Please provide these details.
- Were all samples processed in a single batch or across multiple runs? If the latter, how was batch variability controlled?
- What internal controls or quality checks were in place to monitor bisulfite conversion success?
- Please provide the analytical sensitivity (limit of detection) of the HPV genotyping assay and whether it was validated in-house.
- Lines 127–149: The description of primers and probes is adequate, but no validation data is provided. Were these probes verified for specificity and efficiency?
- Details about assay precision and reproducibility are lacking. Were intra- and inter-assay variability assessed?
- Were ddPCR and qPCR experiments repeated in technical replicates?
- How was the threshold for false-positive methylation signals determined across batches?
- Regarding statistical methods:
- Were adjustments made for multiple comparisons?
- How were ROC thresholds selected (e.g., Youden Index)?
- Were the cutoffs predefined or retrospectively optimised?
- Was internal validation (e.g., cross-validation) performed to assess robustness?
- Table 1:
- (a) Include p-values for intergroup comparisons using chi-square or non-parametric tests.
- (b) Key confounders (e.g., HIV status, STIs, contraceptive use) are not reported—were these data collected?
- (c) Disparities in group sizes could bias comparisons. Were these imbalances adjusted for in the analysis?
- (d) Age differences, especially with older carcinoma patients, could affect methylation. Was age used as a covariate?
- (e) Consider reporting continuous variables like sexual activity duration and partner count using means/SD or medians/IQRs for consistency.
Results
- Figure 1 shows comparable DNA input across groups. However, exact p-values and complete data distributions should be reported to support this statement.
- Were DNA quality and bisulfite conversion success verified before methylation analysis?
- The marked discrepancy between ddPCR and qPCR detection for CADM1 and PAX1 raises concerns. Were qPCR assays optimised before comparison? Were LODs assessed?
- Were ddPCR and qPCR results reproducible? Please consider reporting CVs or standard deviations across replicates.
- CADM1 methylation is significantly associated with HPV-positive status, but similar findings were not observed for MAL and PAX1. Please discuss possible reasons for this discrepancy.
- Did the authors use multivariate models (e.g., logistic regression) to adjust for confounders such as age and sexual history?
- Were any methylation-positive samples found among HPV-negative controls? How were potential false positives managed?
- Table 2 suggests limited discriminative ability of the markers for HSIL. How do the authors interpret this?
- Was combined or co-methylation analysis (e.g., CADM1 + PAX1) performed to assess potential diagnostic synergy?
- Figures 6–7 and Tables S5–S6: While the combined CADM1/MAL panel performs well for carcinoma, its lower sensitivity for HSIL limits its triage value. Were ROC thresholds pre-specified or selected post hoc?
- Was cross-validation or bootstrapping applied to test the stability of ROC metrics?
- Were PPVs and NPVs calculated? These would better indicate real-world clinical performance, particularly in low-prevalence settings.
- Several figures, especially boxplots, lack consistent axis labelling and statistical detail. Standardising figure formatting and adding sample sizes would enhance interpretability.
Discussion
- The biological relevance of the methylated genes is not discussed in depth. Please elaborate on how CADM1, MAL, and PAX1 function in cervical carcinogenesis, and whether HPV oncoproteins directly regulate them.
- Clarify why CADM1 and MAL methylation appear mainly in carcinoma, while PAX1 is seen earlier. Could this reflect their roles in tumour progression stages?
- Consider whether PAX1 methylation might represent broader field cancerisation or non-specific HPV-related changes, potentially explaining its lower specificity.
- How does this marker panel compare to other established triage methods (e.g., cytology, dual staining, and HPV genotyping)?
- What are the clinical implications of high specificity but moderate sensitivity (70%)? How might this impact colposcopy referral decisions?
- Discuss the practical limitations of ddPCR in resource-constrained settings. Could qPCR remain the method of choice in large-scale screening?
- Was any analysis conducted on the cost or throughput of ddPCR relative to qPCR?
- What controls were used to ensure bisulfite conversion efficiency per sample?
- Were any samples excluded due to failed conversion, and if so, how many?
- While the discussion references relevant international studies, it misses an opportunity to position this work as the first ddPCR-based methylation analysis of CADM1/MAL/PAX1 in a Russian population.
- The authors acknowledge the sample size and single-centre design but should also mention the lack of multivariate adjustment, age imbalance, and moderate performance for HSIL.
- Is there a plan to validate this panel in a larger, multicenter cohort with better demographic and clinical matching?
The conclusion summarises key results well.
- Please clarify that the methylation markers perform better for carcinoma than HSIL, limiting their early-stage utility.
- The statement regarding PAX1’s early role may overstate its discriminatory capacity. Please consider tempering this claim or supporting it with additional data.
- The conclusion correctly states the triage value of these markers. Can the authors explain how this approach might help reduce unnecessary colposcopy referrals in HPV-positive women?
- Could the authors recommend future directions, such as portable ddPCR or alternative methylation detection technologies better suited for screening settings?
- More specificity would help future research. Consider including diverse HPV genotypes, more early-stage lesion types, and additional candidate genes.
References
- The text suggests adding clinical guidelines, citing alternative biomarkers, updating outdated citations with more recent literature, and including a recent systematic review on methylation markers in cervical screening to address the unmet need for better triage tools
The manuscript is well-written with clear scientific language and technical vocabulary. However, it requires more concise phrasing, better transitions, and grammatical consistency. A thorough language edit by a professional is recommended.
Author Response
To the Editor-in-Chief of the Biomedicines journal,
We are pleased to submit the revised version of our manuscript entitled:
“The Significance of Genomic DNA Methylation in Identifying Cytological and Histological Abnormalities of the Cervix in Women with High-Risk Human Papillomavirus Infection” (Manuscript ID: biomedicines-3649339).
The comments from the third reviewer were received on May 23, 2025. We have already started working on the detailed response; however, due to the substantial number of points raised (more than 60), we were unable to complete the full response within the current timeframe. Therefore, we kindly request an extension of the revision deadline to allow us sufficient time to prepare a comprehensive and thoughtful response.
Thank you for your understanding and consideration. We look forward to your reply.
Sincerely,
Maria Anisimova, on behalf of all co-authors
Round 2
Reviewer 3 Report
Comments and Suggestions for Authors
I appreciate the authors' thoughtful and substantial revisions in response to the initial review, and I would like to recommend creating a response file to track the comments and suggestions.
That said, a few essential methodological and statistical concerns remain, which should be addressed before the manuscript is considered for publication:
Major Points Still Requiring Revision:
Multivariate Analysis of Confounders: While the demographic table includes potential confounders such as age, sexual history, smoking, and parity, no multivariate statistical analysis (e.g., logistic regression) was performed to adjust their effects. This is particularly important given the observed age differences between diagnostic groups, which could independently influence methylation patterns.
Assay Validation and Reproducibility: The manuscript still lacks information on technical replicates, inter-assay variability, and internal controls for bisulfite conversion. Reporting these elements is essential to establish the reliability and reproducibility of the ddPCR findings.
Statistical Robustness: Although the ROC curve analysis is now clearly presented, cross-validation, bootstrapping, or adjustments for multiple comparisons are not mentioned. These steps would strengthen the credibility of the diagnostic performance claims, especially in a small, single-centre cohort.
Clarification of False Positives and qPCR Performance: The manuscript could benefit from a more straightforward explanation regarding detecting methylation in HPV-negative controls, if any, and how false positives were defined or excluded. Additionally, while the comparative performance of qPCR versus ddPCR is well described, optimisation steps for qPCR before comparison are not fully detailed.
Title Suggestion: To better reflect the study's novelty, I recommend amending the title to include either the specific biomarkers (CADM1, MAL, PAX1) or the mention of ddPCR technology.
I commend the authors for their diligent effort in strengthening the manuscript and encourage them to address the above points to improve its scientific rigour and translational applicability. Once these final revisions are incorporated, the manuscript may offer meaningful contributions to molecular diagnostics in cervical cancer screening.
Author Response
Dated: 03.06.2025
Subject: Submission of Revised Manuscript
To the Editor of the Biomedicines Journal,
We thank the Editor and the reviewers for the opportunity to revise our manuscript titled “The Significance of Genomic DNA Methylation in Identifying Cytological and Histological Abnormalities of the Cervix in Women with High-Risk Human Papillomavirus Infection” (Manuscript ID: biomedicines-3649339). We appreciate the valuable feedback provided and the constructive comments, which have helped us to improve the quality and clarity of our work.
We have carefully revised the manuscript in accordance with the reviewer’s suggestions. All changes have been clearly indicated in the revised version using the track changes function in Microsoft Word.
Below, we provide a detailed, point-by-point response to the reviewer’s comments, structured according to the thematic blocks identified by the reviewer.
- Multivariate Analysis of Confounders
Reviewer’s comment:
“Multivariate Analysis of Confounders: While the demographic table includes potential confounders such as age, sexual history, smoking, and parity, no multivariate statistical analysis (e.g., logistic regression) was performed to adjust their effects. This is particularly important given the observed age differences between diagnostic groups, which could independently influence methylation patterns.”
Response:
Thank you for this valuable comment. In response, we have performed a multivariate logistic regression analysis to evaluate the association between methylation markers (CADM1, MAL, PAX1) and high-grade cervical lesions (HSIL+), adjusting for potential confounders, including age, smoking, parity, and the number of sexual partners.
The analysis demonstrated that CADM1 methylation remained an independent and statistically significant predictor of HSIL+ (OR = 39.546; 95% CI: 1.130–1383.704; p = 0.043). In contrast, MAL (OR = 1.981; p = 0.632) and PAX1 (OR = 1.557; p = 0.479) methylation did not reach statistical significance after adjustment.
Additionally, we performed a Pearson’s correlation analysis to assess the relationships between methylation markers and demographic/clinical variables. The results demonstrated no strong correlations (|r| < 0.4) between methylation levels and factors such as age, parity, smoking, or the number of sexual partners. The correlation matrix has been included as Supplementary Figure S6.
Therefore, CADM1, MAL, and PAX1 methylation levels can be considered independent of the assessed demographic and clinical factors in our study cohort. These results have been incorporated into the revised manuscript (Results section, page 9, lines 315–322).
"Multivariate logistic regression analysis demonstrated that CADM1 methylation was an independent predictor of high-grade cervical lesions (HSIL+) after adjusting for age, smoking, parity, and the number of sexual partners (OR = 39.546; 95% CI: 1.130–1383.704; p = 0.043). However, MAL (OR = 1.981; p = 0.632) and PAX1 (OR = 1.557; p = 0.479) methylation did not reach statistical significance in the model. Correlation analysis showed no strong associations between methylation levels and clinical-demographic factors (age, parity, smoking, number of sexual partners). The correlation matrix is provided in Supplementary Figure S6."
- Assay Validation and Reproducibility
Reviewer’s comment:
“The manuscript still lacks information on technical replicates, inter-assay variability, and internal controls for bisulfite conversion. Reporting these elements is essential to establish the reliability and reproducibility of the ddPCR findings.”
Response:
We thank the reviewer for this valuable comment. We have revised the Methods section to provide detailed information on technical replicates, inter-assay variability, and internal controls for bisulfite conversion.
Regarding technical replicates, the Methods section (page 4, lines 161–163) states: “Samples were analyzed in a single measurement for both ddPCR and qPCR. If the results for a sample did not meet the quality criteria, a repeat reaction was performed.”
For internal controls for bisulfite conversion, the following information is provided in the Methods (page 4, lines 122–125): “The completeness of bisulfite conversion of DNA was verified by a qPCR reaction using oligonucleotides designed to amplify a specific non-converted fragment of the ACTB gene (Table S2, Supplementary Materials).”
For inter-assay variability, the following information is provided in the Methods (page 4, lines 165–169): “As samples were processed across multiple qPCR/ddPCR runs, batch variability was controlled by assessing the reproducibility (deviation ≤10%) of results from a positive control sample containing a 1:1 mixture of Qiagen control methylated DNA and Qiagen control unmethylated DNA (Qiagen GmbH, Hilden, Germany).”
These clarifications have been incorporated into the Methods section and are highlighted in the revised manuscript.
- Statistical Robustness
Reviewer’s comment:
“Statistical Robustness: Although the ROC curve analysis is now clearly presented, cross-validation, bootstrapping, or adjustments for multiple comparisons are not mentioned. These steps would strengthen the credibility of the diagnostic performance claims, especially in a small, single-centre cohort.”
Response:
We thank the reviewer for this important comment. We have revised the Methods section to clarify that the optimal cutoffs for ROC-curve analysis were determined using the Youden index. Specifically, the following sentence was added (page 5, lines 194–195): “The Youden index was used to determine the optimal threshold values during ROC-curve analysis.”
Regarding adjustments for multiple comparisons, we clarified that no such corrections were applied because the study did not involve simultaneous comparisons of multiple groups. This clarification has been added to the Methods section (page 5, lines 195–197): “No adjustments for multiple comparisons were performed, as no simultaneous comparisons of multiple groups were conducted.”
As for internal validation, we clarified that the cutoffs were retrospectively optimized, but due to the limited sample size, no cross-validation or bootstrapping was performed. This information is now included in the Limitations section (page 12, lines 431–433): “Due to the low sample size, internal validation (e.g., cross-validation) was not implemented. The cutoff values for the presented assays require further verification in a larger cohort.”
These clarifications have been incorporated into the revised manuscript and are highlighted using the track changes function.
- Clarification of False Positives and qPCR Performance
Reviewer’s comment:
“The manuscript could benefit from a more straightforward explanation regarding detecting methylation in HPV-negative controls, if any, and how false positives were defined or excluded. Additionally, while the
comparative performance of qPCR versus ddPCR is well described, optimisation steps for qPCR before comparison are not fully detailed.”
Response:
Thank you for this important comment. In response, we have revised the Methods section to provide a more detailed explanation of how false-positive methylation signals were defined and excluded, particularly in HPV-negative controls, and to clarify the optimisation procedures for qPCR.
We added the following sentences to the Methods section: "Analytical sensitivity and specificity of the assays described above were validated in a series of experiments using Qiagen control methylated and unmethylated DNA (Qiagen GmbH, Hilden, Germany), mixed with methylated DNA fractions ranging from 0% to 100%. The limit of detection for qPCR assays was 10% for PAX1 and 15% for CADM1 and MAL; for ddPCR assays, the limit of detection was below 1% (exact determination was impeded by the unreliability of further dilution of methylated DNA). Samples were analyzed in a single measurement for both ddPCR and qPCR. If the results for a sample did not meet the quality criteria, a repeat reaction was performed."
To define and control for false-positive signals, the following methodological detail was added: "False-positive signals across batches were determined (and subtracted from the results of the corresponding samples) based on the threshold cycle or the number of methylation-positive droplets detected in a sample containing Qiagen control unmethylated DNA (Qiagen GmbH, Hilden, Germany)."
Additionally, to further clarify quality control and ensure comparability between assays, the following criteria were described: "The qPCR quality criteria included an ACTB (internal control) fluorescence threshold cycle of ≤32; the ddPCR quality criteria included the generation of >10,000 droplets per well and an ACTB level of >100 copies per well. As samples were processed across multiple qPCR/ddPCR runs, batch variability was controlled by assessing the reproducibility (deviation ≤10%) of results from a positive control sample containing a 1:1 mixture of Qiagen control methylated DNA and Qiagen control unmethylated DNA (Qiagen GmbH, Hilden, Germany)."
These additions clarify how background methylation signals were accounted for, how positive/negative controls were incorporated, and how performance characteristics of the assays were evaluated and standardised prior to comparative analysis. The revised text appears in the Methods section (pages 4-5, lines 155–172).
- Title Suggestion
Reviewer’s comment:
“To better reflect the study's novelty, I recommend amending the title to include either the specific biomarkers (CADM1, MAL, PAX1) or the mention of ddPCR technology.”
Response:
We thank the reviewer for this helpful suggestion. We agree that the title should clearly reflect the biomarkers, the ddPCR method, and the clinical context. In response, we have revised the title as follows: "Detection of CADM1, MAL, and PAX1 Methylation by ddPCR for Triage of HPV-Positive Cervical Lesions."
We believe this revised title is concise, accurate, and aligned with the journal’s style.

Round 3
Reviewer 3 Report
Comments and Suggestions for Authors
Thank you for your thoughtful and detailed responses to the reviewer comments. The revisions you've implemented have substantially strengthened the manuscript. In particular, the addition of multivariate logistic regression enhances the scientific rigour and accounts for potential confounding. Technical details regarding assay optimisation, reproducibility, and bisulfite conversion controls are now clearly presented. The revised title and abstract now more accurately reflect the study's scope, novelty, and clinical utility. Visual data presentation (figures and supplementary tables) has been improved for clarity and interpretability. Your acknowledgement of limitations, such as the lack of cross-validation and moderate sensitivity for HSIL, is appreciated and helps to explain the findings realistically.
I suggest the professional editing service for refinement and minor grammatical corrections, and ensure that figure axes and labels are uniformly formatted across all visual elements.
Overall, this is a well-conducted and timely study with meaningful implications for cervical cancer screening and triage. I recommend acceptance following language refinement and formatting polish. I look forward to seeing it published.
Overall, the manuscript is well-organised. However, it does have recurring grammatical errors, including redundant phrasing, misuse of articles (e.g., using "the" with non-count nouns), inconsistencies in subject-verb agreement, run-on or overly complex sentences, and non-standard punctuation use.
Author Response
Dated: 9 June 2025
Subject: Submission of revised manuscript
To the Editor-in-Chief of the Biomedicines journal,
We are pleased to submit the revised version of our manuscript entitled:
“Detection of CADM1, MAL, and PAX1 Methylation by ddPCR for Triage of HPV-Positive Cervical Lesions” (Manuscript ID: biomedicines-3649339).
We sincerely thank the reviewer for the thoughtful and detailed assessment of our work. We are grateful for the recognition of the improvements made in response to the previous round of comments and for the constructive suggestions that have helped us further refine the manuscript.
In line with the reviewer’s recommendation, we have harmonised the formatting of all visual elements. Specifically, figure axes and labels have been standardized to ensure consistent style, readability, and professional appearance throughout the manuscript and Supplementary Materials.
Additionally, we have carried out a comprehensive language revision of the manuscript. To ensure clarity and eliminate grammatical inconsistencies, we sought assistance from a professional translator and linguist experienced in academic scientific writing.
All revised sections are highlighted using the track changes function.
Thank you for your consideration. We hope that the revised manuscript meets the journal’s standards and look forward to your decision.
Sincerely,
Maria Anisimova, on behalf of all co-authors
Corresponding author:
M. Anisimova
Email: manecha35@mail.ru
Phone: +7-906-018-29-11
